



# Assessment of surface solar irradiance derived from real-time modelling techniques and verification with ground-based measurements

Panagiotis G. Kosmopoulos[1,2], Stelios Kazadzis[3,1], Michael Taylor[4], Panagiotis I. Raptis[1,3], Iphigenia
Keramitsoglou[5], Chris Kiranoudis[5,6], Alkiviadis F. Bais[2]

[1]Institute for Environmental Research and Sustainable Development, National Observatory of Athens, Greece
[2]Laboratory of Atmospheric Physics, Aristotle University of Thessaloniki, Greece
[3]Physicalisch-Meteorologisches Observatorium Davos, World Radiation Center, Switzerland
[4]Department of Meteorology, University of Reading, United Kingdom
[5]Institute for Astronomy, Astrophysics, Space Applications and Remote Sensing, National Observatory of Athens, Greece
[6]School of Chemical Engineering, National Technical University of Athens, Greece

*Correspondence to:* P.G. Kosmopoulos (pkosmo@meteo.noa.gr)

**Abstract.** This study focuses on the assessment of surface solar radiation (SSR) based on operational Neural Network (NN) and Multi-Regression Function (MRF) modelling techniques that produce instantaneous (in less than one minute) outputs. Using real-time cloud and aerosol optical properties inputs from the Spinning Enhanced Visible and Infrared Imager (SEVIRI) onboard the Meteosat Second Generation (MSG) satellite and the Copernicus Atmosphere Monitoring Service (CAMS), respectively, these models are capable of calculating SSR in high resolution (1 nm, 0.05 degrees, 15 min) that can be used for spectrally-integrated irradiance maps, databases and various applications related with energy exploitation. The real-time models are validated against ground-based measurements of the Baseline Surface Radiation Network (BSRN) in a temporal range varying from 15-min to monthly means, while a sensitivity analysis of the cloud and aerosol effects on SSR is performed to ensure reliability under different sky and climatological conditions. The simulated outputs, compared to their common training dataset created by the radiative transfer model (RTM) libRadtran, showed median error values in the range -15 to 15% for the NN that produces spectral irradiances (NNS), 5-6% underestimation for the integrated NN and close to zero errors for the MRF technique. The verification against BSRN revealed that the real-time calculation uncertainty ranges from -100 to 40 W/m$^2$ and -20 to 20 W/m$^2$, for the 15-min and monthly mean Global Horizontal Irradiance (GHI) averages, respectively, while the accuracy of the input parameters, in terms of aerosol and cloud optical thickness (AOD and COT), and their impact on GHI, was of the order of 10% as compared to the ground-based measurements. The proposed system aims to be utilized through studies and real-time applications, which are related with the solar energy production planning and use.



**Keywords.** Surface Solar Radiation; Real-time; Neural Network; Radiative Transfer Model; Multi-Regression Function; Aerosol; Cloud; CAMS; MSG; BSRN; PFR

## 1 Introduction

Solar energy exploitation is a cornerstone for sustainable development, through efficient energy planning, towards gradual independence from fossil fuels. To this direction, the European Union (EU), the Middle East and North Africa (MENA) and numerous neighbouring regions and countries, have laid out specific technology roadmaps aiming at the integration of low carbon energy technologies linked with the deployment of photovoltaic (PV) installations in the energy market (IPCC, 2012; NREL, 2016; IRENA, 2016; Jager-Waldau, 2016; REN21, 2017; UN, 2017). In addition, the United Nations (2017) have set as main sustainable development goal by 2030, to ensure universal access to affordable, reliable, and modern energy services. The International Energy Agency (2007) has estimated that the global primary energy demand will increase by 40-50% from 2003 to 2030. Since energy production, transportation and consumption put considerable pressure on the environment, there is serious concern regarding the sustainability of energy consumption.

Earth observation (EO) based systems and relevant services already play an important role in the solar energy industry, as well as in human health related emerging technologies, but there is still significant potential in increasing their efficiency and exploitation (Schroedter-Homscheidt et al., 2006; Wald et al., 2011; Lefevre et al., 2014). EO from space is already triggering services and applications that can deliver benefits throughout all the phases of energy production and supply. Their contribution ranges from identifying reservoirs and locations with solar energy potential, to controlling and monitoring of the distribution networks across Europe, Africa and Middle East, while providing support to energy policy formulation and enforcement (EU, 2011; IEA, 2010).

The need for improved EO-based surface solar irradiance assessment is increasing as more solar farms are included in national electricity grids, worldwide (EC, 2013). Solar energy related installations have been increasing their share on the total energy demand as defined by the Distribution and Transmission System Operators (DSOs and TSOs, respectively). As a result, accurate, real-time and short-term forecasting estimations of the surface solar radiation (SSR) and more specifically the global horizontal irradiance (GHI) related with the operation principles of PV installations, are vital. The real time GHI estimations are required at local and regional scales, and high temporal frequency (every 5-15 minutes), in order to be used for near real time decisions, linked with the PV related contribution to the electricity grid.

Since the launch of EO satellites, such as Meteosat Second Generation (MSG), and Sentinel satellite series, real time image processing techniques have been developed (Suárez and Nesmachnow, 2012). The main advantage of these techniques is the possibility to monitor numerous meteorological variables in almost real time (Derrien et al., 2005; MeteoFrance, 2013). A comprehensive intercomparison of radiation products, codes, algorithms, models and independent data banks has been performed by many researchers (Oreopoulos et al., 2010; 2012; Ellingson et al., 1991; Ineichen, 2006; Beyer et al., 2009; Cahalan, et al., 2005). Solid steps in estimating the surface GHI were taken by Deneke et al. (2008), Schulz et al. (2009),



Mueller et al. (2009), Huang et al. (2011) and Qu et al. (2017), who developed GHI retrieval methodologies based on the use of discrete pre-calculated look-up tables (LUT), while Dorvlo et al. (2002), Zarzalejo et al. (2005), Lopez et al. (2001) and Takenaka et al. (2011) developed solutions based on neural network (NN) models. The validation of most of the above mentioned methodologies was performed against radiative transfer model (RTM) simulations and ground-based

measurements, from various networks around the globe. However, from the validation results it was highlighted that accuracy was inversely proportional to calculation speed under all sky and terrain conditions. The magnitude of the GHI uncertainty due to the effect of aerosols and clouds is significant and has motivated numerous related studies (Federico et al., 2017; Kosmopoulos et al., 2015; Lara-Fanego et al., 2012; Tegen et al., 1996; Lindfors et al., 2013). Under high aerosol loads the SSR can be reduced by 20-50% (Eck et al., 1998; Gleeson et al., 2016; Kosmopoulos et al. 2017), while under

cloudy conditions the impact was up to 60-90% for overcast conditions and cloud coverege of 8 octas (Aebi, et al., 2017; Kosmopoulos et al., 2015; Zygmuntowska et al., 2012), highlighting the significant effect of these atmospheric parameters (clouds and aerosols) on the GHI calculations and in the performance of PV installations and energy production.

In the present study, we report on (i) the assessment of the surface solar irradiance calculated in real-time; which is defined as the product with a time delay of one minute or less from an actual atmospheric situation, by developing and using two

NN-based techniques and a multi-regression-function-based technique and (ii) the validation of these techniques against ground-based measurements from the Baseline Surface Radiation Network (BSRN). Section 2 presents data, methods and techniques used. Section 3 describes the validation results including a sensitivity analysis of related atmospheric parameters and in Section 4 we present our conclusions on the proposed techniques.

## 2 Data and Methodology

### 2.1 Data

### 2.1.1 Ground-based measurements

The verification of the applied SSR real-time modelling techniques was performed against ground-based measurements from nine stations (Table 1) of the Baseline Surface Radiation Network (BSRN) (Hegner et al., 1998) equipped with Kipp and Zonen pyranometers (GHI measurements) and a Precision Filter Radiometer (PFR) at Izaña, Spain. BSRN consists of high-

quality ground based measurements of SSR and for the purposes of the comparison we used the dataset from July 2014 to June 2015. Table 1 presents the location and description of the nine BSRN stations used for the validation of the SSR estimations calculated with the modelling techniques. The temporal resolution of the ground-based measurements is 1 minute, so in order to match the 15-min resolution of the MSG cloud data (and hence the SSR outputs) we used 15-min averages of all the BSRN and PFR measurements used. The selected BSRN stations represent a variety of different climates,

altitudes and aerosol sources in the field of view of MSG and thus provide an opportunity to study the models performance under various atmospheric conditions.



### 2.1.2 Real-time cloud observations

The most important input to our real-time modelling techniques, were the satellite cloud data products from the Spinning Enhanced Visible and Infrared Imager (SEVIRI) onboard the Meteosat Second Generation (MSG) satellite. We obtained the cloud type (CT), the cloud phase (CP) and the cloud optical thickness (COT) products, as to efficiently quantify the effect of clouds on SSR. COT depends on the moisture density as well as the vertical thickness of the cloud. The cloud reflectance at

channel at 0.6 μm in the visible part of the electromagnetic spectrum is directly related with COT (Roebeling et al., 2006). MSG geostationary satellite, because of its orbit height (36,000 km above the equator) allows the continuous monitoring of the area over Europe, Africa and partly South America at high temporal and spatial resolution (15 minutes and $0.05^o$, respectively). The operational MSG-SEVIRI data were acquired by the EUMETCast station operated by the Institute for

Astronomy, Astrophysics, Space Applications and Remote Sensing of the National Observatory of Athens. The cloud properties are extracted operationally and in real-time using the Satellite Application Facility for Nowcasting Weather Conditions software (SAFNWC) installed in-house. CT and CP are standard output products of the SAFNWC computational procedure, while COT is a tailor-made product and as a result its extraction required an additional intervention in the process chain. The cloud product identification is described in Derrien and Gléau (2005) and MeteoFrance technical report (2013). In

the current implementation, cloud products are provided operationally for the entire Earth disk view area of MSG. We extracted products at specific pixels corresponding to locations of the BSRN stations, that were used as inputs to the SSR modeling techniques.

### 2.1.3 Aerosol forecasts

For the real-time assessment of the SSR we additionally incorporated as basic input parameter the aerosol 1 day forecast data

from the Copernicus Atmospheric Monitoring Service (CAMS). These forecasts are based on the Monitoring Atmospheric Composition and Climate (MACC) reanalysis tools, and include validated modelling of aerosol and satellite data assimilation (Eskes et al., 2015). They are able to provide operationally accurate data of aerosol optical depth (AOD) at 550 nm, at 1 hour time steps and $0.4^o$ spatial resolution. The estimation of the aerosol sources is extracted from the Emission Database for Global Atmospheric Research (EDGAR) and the Speciated Particulate Emission Wizard (SPEW), while the

reliability of the product is supported by continuous assimilation into the model of the MODIS AOD data, applying a bias correction from multiple data sources (Dee and Uppala, 2009). For the purposes of our SSR estimations, the CAMS AOD forecasts with the MSG COT data described above, constitute the most important input parameters, together with solar elevation, for the SSR retrieval modelling tools.

### 2.2 Methodology

In this section we present the SSR real-time modelling techniques, the methodology used for developing operational products and the validation statistics against ground-based measurements. The techniques are the Multi-Regression



Functions (MRF), the Neural Network that produces spectral irradiances (NNS) and which is presented in detail in Taylor et al. (2015), and a variant version of the NN that produces integrated irradiances. All three techniques have been optimized based on LUTs that are described in the section 2.2.1 and produce instantaneous (with less than one minute delay from the time that the MSG image is produced) SSR. The number of outputs depends on the region under study and can be of the

order of $10^6$ simulations, simultaneously. In this study we used as operational inputs the CAMS AOD and the MSG COT, in conjunction with the solar elevation angle, as they are the major attenuators of the GHI. Since the comparison of real-time modelling techniques with ground-based measurements are performed from southern Africa to northern Europe, the verification will be focused on GHI. Utilization of the Direct Normal Irradiance (DNI) by Concentrated Solar Power (CSP) plant installations is limited at places with high amounts of DNI (Green et al., 2015) and hence CSPs are de facto outside

energy planning for the majority of the counties represented by the nine BSRN stations and the MSG view. Figure 1 illustrates the procedural flows of the three developed real-time modelling techniques for operational use. Starting with the MSG cloud flags (0=clear sky and 1, 2, 3=cloudy sky in terms of water, ice and mixed clouds, respectively), we identify the clear-sky and cloudy-sky pixels. For the cloudy pixels we incorporate the optical properties (COT) and types of clouds (CT), while for clear sky pixels we take into account the aerosols effect (AOD) and the total ozone column (TOC), which was

derived using Ozone Monitoring Instrument (OMI) retrievals (Wandji-Nyamsi et al., 2015). Then, for all sky conditions we generate the input files to the real-time techniques and, depending on their special characteristics, we produce spectral or spectrally weighted products (see following sub-sections) at high spectral, spatial and temporal resolution (1 nm, 0.05°, 15 min). The actual outputs can be SSR time series, local and regional maps or Earth disk view maps (Fig. 2).

The performance of the real-time techniques was evaluated by comparing the GHI outputs with (i) the initial RTM

simulation LUTs, (ii) the BSRN ground-based measurements and with respect to the aerosol and cloud effects. The evaluation was based on the bias and mean bias error (MBE), the root mean square error (RMSE) and their relative components (rMBE and rRMSE, respectively):

$$MBE = \bar{\varepsilon} = \frac{1}{N}\sum_{i=1}^{N}\varepsilon_i \qquad (1) \qquad\qquad RMSE = \sqrt{\frac{1}{N}\sum_{i=1}^{N}\varepsilon_i^2} \qquad (2)$$

The residuals (estimation errors), $\varepsilon_i = x_e - x_0$, are calculated as the difference between the estimated values by the real-time techniques ($x_e$) and the measured values ($x_0$) by BSRN, where N is the total number of data. MBE measures the overall bias

and detects the model's overestimation (MBE>0) or underestimation (MBE<0). RMSE quantifies the spread in the distribution of errors. In addition, for the various tests performed in this study we calculated the slope, the correlation coefficient (r), the percentage difference (%), the mean absolute difference and the standard deviation.

**2.2.1 Radiative Transfer Model**

All modelling techniques presented in this paper for the real-time assessment of the SSR, are based on LUTs, calculated with

the radiative transfer model (RTM) libRadtran (Mayer and Kylling, 2005; Emde et al., 2016). These LUTs are described in



detail in Taylor et al. (2015) and consist of more than 2.5 million RTM simulations with atmospheric inputs and 1 nm spectral resolution GHI outputs. The interoperable exchange of similar GHI databases is studied by Ménard et al. (2015) highlighting the usefulness and necessity of such LUT-based approaches (Lefevre et al., 2014). Under clear-sky conditions the simulated by libRadtran input parameters were the solar zenith angle (SZA), the AOD, the Ångstrom exponent (AE), the

single scattering albedo (SSA), TOC and the columnar water vapour (WV), while under cloudy conditions except from SZA and TOC, we also used the optical thicknesses of water and ice clouds (WCOT and ICOT, respectively) as inputs. The AOD is not used for cloudy conditions when COT>1, as the effects of aerosols are much weaker compared to thick clouds. For the model versus BSRN station comparison, in order to take into account the station altitude, an altitude correction on the solar energy output of the different model simulations has been applied based on RTM (Libradtran) calculations. The outputs are

high resolution spectral irradiances (1 nm) covering the wavelength region between 285 and 2700 nm. In brief, we used the SDISORT radiative transfer solver (Dahlback and Stamnes, 1991) with pseudospherical approximation to produce valid outputs from 0 to 90$^o$ SZA; the simulations were calculated using a band parameterization method based on the correlated K-approximation (Kato et al., 1999), while the aerosol and cloud determination was performed based on the default aerosol model described by Shettle (1989) and typical cases for the height of water and ice clouds, the effective radius ($R_{eff}$) and the

liquid water path (Hess et al., 1998). All the technical and structural information about the RTM simulations, the input parameters and the construction of the LUTs is presented in Taylor et al. (2015). Table 2 presents the slope and the correlation coefficient between the RTM simulations of GHI and the BSRN ground-based measurements for the whole datasets and period. The overall accuracy in terms of slope ranges from 0.866 (CAB) to almost 1 (0.999 at TOR), while the r values range between 0.93 and 0.97.

## 2.2.2 Multi-Regression Function

The multi-regression function (MRF) technique was developed as an analytical methodology using the RTM outputs, with the aim to provide results as close as possible to the initial (training set) RTM outputs. The advantage in the use of these functions is that they can be executed very rapidly and can be used for real-time SSR determination. In order to achieve that, analytical functions for the SSR should be constructed. In general, SSR is a function of SZA, COT, AOD, AE, SSA, WV and

TOC. For AE and SSA we used monthly climatological values in order to bridge the gap between the operational input availability and the SSR accuracy. However, a preliminary investigation has been performed for the sensitivity of GHI to WV column and TOC. We compared integrated spectral GHI over the entire spectrum for different TOC values and we found a mean difference of only 0.5% for TOC ranging between 300 and 400 DU. For WV columns ranging between 0.5 and 2 cm we found a mean difference of 3.2%, although for SZA<15$^o$ this difference was higher, up to 5%. So, we chose to

neglect these variables in the first place and use TOC=350 DU and WV=0.5 cm for further calculations, considering the differences mentioned above as a scale of error introduced by this approach.

Then, we constructed different polynomial functions according to Gasca and Sauer (2000) for cloudy and clear-sky conditions, to be applied into the scheme presented in Fig. 1. For cloudy cases the irradiance is expressed as f_cloud(SZA,





COT) and for clear-sky cases as f_clear(SZA, AOD). We tested different orders of two-variable polynomials to conclude on the best regression (multi-regression analysis) and we found that the estimates closest to the RTM results were achieved using 5th and 4th polynomials (Sauer and Xu, 1995), as follows:

$$f(x,y) = p_{00} + p_{10}x + p_{01}y + p_{20}x^2 + p_{11}xy + p_{02}y^2 + p_{30}x^3 + p_{21}x^2y + p_{12}xy^2 + p_{03}y^3 + p_{40}x^4 + p_{31}x^3y$$
$$+ p_{22}x^2y^2 + p_{13}xy^3 + p_{04}y^4 + p_{41}x^4y + p_{32}x^3y^2 + p_{23}x^2y^3 + p_{14}xy^4 + p_{05}y^5 \quad (3)$$

where x is SZA and y is AOD and COT accordingly (clear or cloudy sky pixels). Table 3 presents the analytical values of $p_{xx}$ for the purposes of this study (GHI) under clear and cloudy sky conditions. By this approach RTM simulations of SSR are derived in computational times that can be applied in any real-time application.

### 2.2.3 Neural Network

As presented in Taylor et al. (2015), the LUT approach, despite its large size, still provides estimates at discrete input values. The interpolation techniques to correct the input-output parameter intervals are computationally more costly than a continuous function-approximating model, or a NN model, which is more preferable for producing real-time outputs (Hornik et al., 1989). Hence, based on the developed LUT, we trained two sets of NNs, each one consisting of a clear-sky and a cloudy-sky specific NN. For multivariate input-output data, feed-forward NN having a minimum of one layer of "hidden"
neurons whose activation functions are nonlinear hyperbolic tangent functions or other general nonlinear sigmoidal functions, has been shown in the literature to be a universal function approximator (Cybenko, 1989; Hornik et al., 1989). The input-output vectors used in this study were connected via two network layers - the first containing hidden neurons with Tanh activation functions and the second containing output neurons with linear activation functions. The exact mathematical equation relating the NN outputs to the NN inputs for this type of NN is given in the following matrix equation described
analytically in Taylor et al. (2014):

$$Y = f^2(LW^{2,1}f^1(IW^{1,1}X + b^1) + b^2) \quad (4)$$

The multiplication of the matrix IW[1,1] and the vector X is a dot product equivalent to the summation of all input connections to each neuron in the hidden layer. This equation is the continuous and nonlinear functional approximation that relates the
output vector to the input vector. This NN approach, its training procedure and all the technical details are described analytically in Taylor et al. (2015).

In the first NN set, we produced instantaneous SSR spectra of the order of 1 million in less than 1 minute, using as operational inputs the CAMS AOD one-day forecasts, the MSG COT and real-time calculations of SZA. The output resolution is high in terms of spectral (1nm), spatial (0.05° degrees) and temporal (15 min) components (Taylor et al., 2015),
and, operational speaking, this spectral-based NN (NNS) can incorporate additional inputs as described in Section 2.2.1.





Similar studies on the temporal variability of SSR by means of spectral representations and the wavelengths absorption parameterization applied to satellite channels and spectral bands were performed by Gasteiger et al. (2014) and Belgulescu et al. (2016). For the purposes of this study we used monthly climatological values for the rest of the input parameters. More specific: TOC from OMI (2007-2016), WV from the Medium Resolution Imaging Spectrometer (MERIS) onboard ESA's Environmental Satellite (ENVISAT), and AE and SSA from the AeroCom database (Kinne et al., 2006). The second NN set, was trained using integrated SSR over the whole wavelength range using the LUT's spectral data. The SSR results of this technique (called hereafter NN) are more accurate in terms of GHI, DNI and Diffuse Horizontal Irradiance (DHI), as it will be discussed in the following section. On the other hand, spectrally-weighted products like the UV-index, the Photosynthetically Active Radiation (PAR) or the Vitamin D effective dose (VDED), cannot be produced with this approach, as only the NNS is able to produce the spectral irradiance needed for such applications.

## 3 Results

### 3.1 Performance of real-time techniques

#### 3.1.1 Comparison with RTM

This section initially summarizes the performance of all the real-time modelling techniques against the RTM simulations for all BSRN stations. Figure 3 presents the percentage difference between the RTM simulations and the MRF, NN and NNS techniques. All data presented here are GHI model outputs with a 15 minute temporal resolution. The box-plots represent the inter-quartile range between the 25 and 75 percentiles with the in-box line to show the median and the upper and lower whiskers to represent the maximum and minimum error values that are within 1.5 times the inter-quartile range of the box edges. The largest differences for all techniques occur for LER and TOR stations followed by CAB, indicating higher introduced uncertainties over highest latitudes, as observed on the MSG Earth view edges. However, differences for MRF are much smaller for these three stations. For all stations MRF shows differences around zero, showing a quite efficient representation of the LUT-based RTM simulations. For the altitude correction (described in section 2.2.1) we included a 4.2% per km at $20^{o}$ SZA up to 12% per km at $80^{o}$ corrections (Fig. 4), based on libRadtran model sensitivity analysis.

The NN and NNS approaches showed a systematic underestimation; for the NN of ~8%, while the NNS had comparatively the worst performance with differences in the range -15 to 80% (LER station - for the inter-quartile ranges). The median differences for NNS range from -15 to 15%, for NN are ~5-6% and for MRF are less than 1%. It is obvious that spectral output methods (NNS) provide more detailed information (e.g. for specialized studies on spectral impacts on the yield of different PV technologies) (Dirnberger et al., 2015; Ishii et al., 2013), but they are more uncertain than the NN and MRF that produce integrated SSR.

### 3.1.2 Verification with BSRN





The model accuracy was verified against nine BRSN stations. We calculated the regression of the mean GHI between the ground measurements and the model outputs, shown in Fig. 5. We also show the intra-model regression compared to the initial RTM simulations (Fig. 5 left), in order to assess the NN and NNS included interpolations of the LUT outputs and the MRF performance. We found that the MRF technique presents identical values with the RTM, for all ground stations and

under all climatological conditions. The NN and NNS show a quite good agreement too in terms of absolute values, as under all conditions, mean GHIs are less than 5% different from the BSRN measurements. In Fig. 5 (right) we confirmed the similarity of MRF with RTM and in some cases with the NN models, indicating the overall efficiency of all interpolation and multi-function techniques used. A slightly better performance was observed for higher mean GHIs proving the usefulness under high solar energy potential conditions.

Figure 6 shows the accuracy of MRF, being the most reliable technique as presented in Fig. 3, with respect to the ground-based measurements, for various temporal integrations, starting from the (actual derived) 15-min to hourly, daily and monthly averages. The uncertainty range of the MRF simulations given as mean inter-quartile GHI differences is highest (from -100 to 40 $W/m^2$, depending on the station) for the 15-min resolution. It is reduced for hourly and daily averages (-70 to 40 $W/m^2$ and -40 to 30 $W/m^2$, respectively), and is minimized for the monthly averages (-20 to 20 $W/m^2$). In particular,

IZA and TAM showed the highest differences for all temporal retrievals, while LER and TOR presented minimum differences down to $\pm20$ $W/m^2$ for the inter-quartile range of the 15-min averages. The median values are within 10 $W/m^2$ for the 15-min and hourly resolutions, while the corresponding minimum and maximum error values (represented in Fig. 6 as the upper and lower whiskers) extends from -200 to 100 $W/m^2$ for the aforementioned resolutions and are reduced to $\pm60$ $W/m^2$ and $\pm40$ $W/m^2$ for the daily and monthly averages, respectively. These results are comparable with similar model

verification approaches and studies (Riihela et al., 2015; Muller et al., 2015; Thomas et al., 2016; Eissa et al., 2015a; 2015b). Indicatively, Muller et al. (2015) and Riihela et al. (2015) discussed the CM-SAF SARAH (Solar surfAce RAdiation Heliosat) data record, which are post processed data. They calculated a mean monthly error for GHI of 5.5 $W/m^2$ and a mean daily error of 12.1 $W/m^2$, with additional uncertainties in terms of spatial representativeness and measurements quality of about $\pm12$ $W/m^2$, while they did not provide relevant information about the hourly or even higher time resolution. The

overall accuracy of all models was evaluated also with respect to seasonality. In Fig. 7 we present the seasonal rRMSE values of the GHI estimations produced by the MRF, NN and NNS models as compared to the BSRN 15-min intervals measurements. The rRMSE for MRF, ranges from 5 to 48% for GOB and TOR stations respectively, for NN the range is increasing to 6-60% and for NNS the corresponding range is 7-87%. We need to highlight that the aforementioned large differences correspond to significantly low absolute GHI values indicating the impact under cloudy conditions mainly in the

winter season, at stations with high mean cloudiness (LER, TOR, CAM and CAB). In the summer, results are better for all stations and for all models (5-29%), while GOB, IZA and TAM stations showed the lowest rRMSE values (5-12% in summer and 12-17% in winter), linked with their lower cloudiness. Eissa et al. (2015a; 2015b) validated the HelioClim-3 database and the McClear model in Egypt and in the United Arab Emirates, and they found RMSE of 68.4-151.7 and 22-47 $W/m^2$, respectively (we found a range of 58.2-70.8 $W/m^2$). Thomas et al. (2016) validated the latest version of HelioClim-3



(v5) against BSRN and found rRMSE of 14.1-37.2% for the 15-min averages, which are directly comparable to our 15-min results (12-35.7%). In particular, for the LER, TOR, CAB, CAM, CAR and TAM stations they found rRMSE of 37.2, 33, 29.4, 25.9, 16.3 and 15.8%, while looking to our MRF performance evaluation results we observe 35.7, 35.6, 29.9, 30.3, 20.2 and 12.2% for the same stations. This indicates that the use of the suggested real-time modelling techniques enables the

production of instantaneous, high resolution and quite accurate (as compared to the post-processed databases) GHI outputs that can be used for solar energy related applications and studies. A detailed presentation of results for all metrics and stations can be found in Table 4.

### 3.2 Sensitivity Analysis

### 3.2.1 Cloud effect

The cloud effect via the radiative transfer of solar radiation in the atmosphere, represents the greatest source of uncertainty in the simulation of SSR, while several models do not have the capability to deal with clouds coexisting with a radiatively active atmosphere (Cahalan et al., 2005). Small changes in cloudiness and its optical properties can impact on GHI. The magnitude of the cloud effects on the model to BSRN comparison can be seen in Fig. 8. Under clear-sky conditions (Fig. 8 left plot), the regression of the 15-min modeled GHI values show very good agreement when compared with the BSRN

measurements for both MRF and NN-based techniques. We plotted the RTM simulations as well in order to depict the corresponding regression. The distinct scatter shown under all-sky conditions (Fig. 8, right) with the cloud cases linked with an underestimation of the modeled GHI in comparison to the BSRN values. This effect has to do with the MSG COT uncertainties and hence introduces errors to the outputs of the SSR techniques (Derrien and Le Gléau, 2005; Pfeifroth et al., 2016). In addition, comparison principles of a (point) station GHI measurements with a $0.05^o$ MSG cloud "picture" are

responsible for part of the observed deviations. As an example, for instants that the MSG $0.05^o$ grid is partly cloudy, the BSRN GHI measurements could fluctuate more than 100%, depending on whether the sun is visible or if clouds attenuate the direct component of the solar irradiance. As a result, in the case of partly covered $0.05^o$ pixel and in the absence of clouds between the BSRN instrument and the sun, BSRN measured GHI would be much higher than the modeled one. Of course the opposite situation is feasible as well causing consequently an overestimation of the modeled GHI (Koren et al., 2007).

Figure 9 illustrates the mean percentage difference and standard deviation of the 15-min GHI produced by the MRF and the measured values by the BSRN stations (only instances with cloudy conditions were used for all stations) as a function of COT. For COT< 2, the MRF technique results higher GHI values than those actually measured, of 1-12%, while as the COT values get higher, the MRF underestimates the measurements by up to -60% for COT around 35. We note that under such high COT values the mean radiation values are much lower than 50 W/m$^2$. The standard deviation reaches its highest value

of 43 W/m$^2$ for COT 14-16 while its lower value of 32 W/m$^2$ is found for COT 2.6.

### 3.2.2 Aerosol effect



In addition to the clouds, aerosols play an important role in the solar radiation transfer in the atmosphere. Especially in places with high solar energy potential, where cloud-free conditions prevail during the largest part of the year, significant aerosol sources could exist (Gkikas et al., 2012). The aerosols effect is closely related to the aerosol optical properties and mostly AOD and as a consequence, the uncertainty in the model AOD input could result to significant errors in the

assessment of SSR (Oumbe et al., 2015; Kosmopoulos et al., 2017). For the purposes of this study we used the Global Atmosphere Watch (GAW-PFR) station of IZA, which is an internationally recognized test bed for aerosol remote sensing instruments (Cuevas et al., 2016), to quantify the AOD difference between the operational input from CAMS and a PFR instrument, under high altitude conditions (Garcia et al., 2013). In Fig. 10 we present the yearly frequency distribution of the differences between CAMS and PFR values for cloudless sky conditions. The majority of the AOD differences are lower

than 0.2 with the maximum frequency encountered at zero AOD differences, indicating the overall good accuracy of CAMS-derived one-day forecasts of AOD. The mean absolute difference was found equal to 0.1075±0.1038 (1 sigma). This shows an overestimation of 0.1 for CAMS that could be lead to MRF GHI small underestimation of 2% compared with BSRN measured GHI. Finally, in Fig. 11 a scatterplot of the CAMS-PFR differences in AOD is shown as a function of absolute differences in GHI derived between the MRF technique and the IZA measurements. The GHI differences are spread around

zero independently of the AOD difference showing the negligible dependence of such small AOD differences to the GHI model calculations.

## 4. Summary and conclusions

This study proposed state-of-the-art modelling techniques (NNS, NN, MRF) for the real-time estimation of SSR, which have been validated against ground-based BSRN measurements. The determination and understanding of the input parameter

effects on radiative transfer, revealed that the accuracy of simulations depends on the quality and resolution of the atmospheric inputs to the models (mostly COT and AOD), while increasing the calculation speed and including spectral GHI information, decreases the model accuracy.

We firstly described the developed modelling techniques which are based on large LUTs for clear-sky and cloudy conditions. Verification of these models was performed for the GHI against ground-measurements at nine stations, with

variable geographical, atmospheric and altitudinal conditions. The comparison showed a dependence on seasonal variability, with summer rRMSE values below 30% for all models, under all conditions, and revealed largest errors for the NNS technique because of the spectral special characteristics, as well as for LER and TOR stations. The NN presented a slight underestimation of 8% against its training RTM simulations, while against BSRN stations succeeded MBE and RMSE values lower than 30 and 80 W/m$^2$ respectively for the annual period, indicating relatively good agreement under various

conditions. The technique with the most accurate results, almost identical to the RTM simulations, was the MRF. Under different temporal scales the mean GHI differences in terms of 25[th] to 75[th] inter-quartiles, compared to the nine stations, were




found to range from -100 to 40 W/m$^2$ for the 15-min intervals and -70 to 40 W/m$^2$ for the hourly means, to -40 to 30 and -20 to 20 W/m$^2$ for the daily and monthly averages and almost 10 W/m$^2$ for the median of difference of each station.

The results presented here show the potential use of such techniques for solar energy related applications and electricity grid supporting services (IRENA, 2015). Comparison of the proposed real-time models with existing databases (SARAH, etc), which in most cases are post processed data using past data series, showed similar results. Finally, we tested the impact of cloud and aerosol inputs to the models in order to reveal the AOD forecast accuracy of CAMS, which turned out to be ~0.1 in absolute terms as compared to ground-based sun-photometric measurements. The MSG COT is related with MRF underestimation of the order of 60% under highly cloudy conditions (COT>30) and negligible GHI levels (<50 W/m$^2$). As a result, the presented real-time models based on the synergy of satellite products, RTM and NN or MRF techniques, are a promising tool to be used within the solar energy related community. Improvements on satellite based model inputs from latest and future satellite missions (e.g. Sentinel missions) could be implemented in the future in the existing system in order to improve spatial and temporal resolution and GHI accuracy.

## Acknowledgements

This research has been partly funded by the H2020 GEO-CRADLE project under grant agreement No 690133, the IERSD/NOA's action THESPIA with grant number PDE2013SE01380031 under the call KRIPIS and the project Aristotelis-SOLAR.

## Nomenclature & abbreviations

| | |
|---|---|
| AE | Angstrom Exponent |
| AOD | Aerosol Optical Depth |
| BSRN | Baseline Surface Radiation Network |
| CAMS | Copernicus Atmosphere Monitoring Service |
| CM SAF | Satellite Application Facility on Climate Monitoring |
| COT | Cloud Optical Thickness |
| CP | Cloud Phase |
| CSP | Concentrated Solar Power |
| CT | Cloud Type |
| DHI | Diffuse Horizontal Irradiance |
| DNI | Direct Normal Irradiance |
| DU | Dopson Unit |
| EDGAR | Emission Database for Global Atmospheric Research |
| ENVISAT | Environmental Satellite |
| EO | Earth Observation |
| ESA | European Space Agency |
| EU | European Union |
| GAW | Global Atmosphere Watch |
| GHI | Global Horizontal Irradiance |



| ICOT | Ice Cloud Optical Thickness |
|---|---|
| libRadtran | Library for Radiative transfer |
| LUT | Look Up Table |
| MACC | Monitoring Atmospheric Compsition and Climate |
| MBE | Mean Bias Error |
| MENA | Middle East and North Africa countries |
| MERIS | Medium Resolution Imaging Spectrometer |
| MODIS | Moderate Resolution Imaging Spectroradiometer |
| MRF | Multi-Regression Function |
| MSG | Meteosat Second Generation |
| NN | Neural Network |
| NNS | Neural Network Spectral |
| OMI | Ozone Monitoring Instrument |
| PAR | Photosynthetically Active Radiation |
| PFR | Precision Filter Radiometer |
| PV | Photovoltaics |
| rMBE | relative Mean Bias Error |
| RMSE | Root Mean Square Error |
| rRMSE | Relative Root Mean Square Error |
| RTM | Radiative Transfer Model |
| SAFNWC | Satellite Application Facilities for NoWCasting |
| SARAH | Solar surfAce RAdiation Heliosat |
| SEVIRI | Spinning Enhanced Visible and InfRared Imager |
| SPEW | Speciated Particulate Emission Wizard |
| SSA | Single Scattering Albedo |
| SSR | Surface Solar Radiation |
| SZA | Solar Zenith Angle |
| TOC | Total Ozone Column |
| UV | Ultraviolet |
| VDED | Vitamin D Effective Dose |
| WCOT | Water Cloud Optical Thickness |
| WV | Water Vapour |

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



**Table 1:** Coordinates (degrees) and height (meters above sea level) of the BSRN stations used for the validation

| Station | Country | C ode | Latitude (ºN) | Longitude (ºE) | Height (m.a.s.l.) |
|---|---|---|---|---|---|
| Gobabeb | Namib Desert, Namibia | **GOB** | -23.5614 | 15.0420 | 407 |
| Izaña | Tenerife, Spain | **IZA** | 28.3094 | -16.4993 | 2373 |
| Tamanrasset | Algeria | **TAM** | 22.7903 | 5.5292 | 1385 |
| Cabauw | Netherlands | **CAB** | 51.9711 | 4.9267 | 0 |
| Camborne | United Kingdom | **CAM** | 50.2167 | -5.3167 | 88 |
| Carpentras | France | **CAR** | 44.0830 | 5.0590 | 100 |
| Cener | Spain | **CNR** | 42.8160 | -1.6010 | 471 |
| Lerwick | United Kingdom | **LER** | 60.1389 | -1.1847 | 80 |
| Toravere | Estonia | **TOR** | 58.2540 | 26.4620 | 70 |






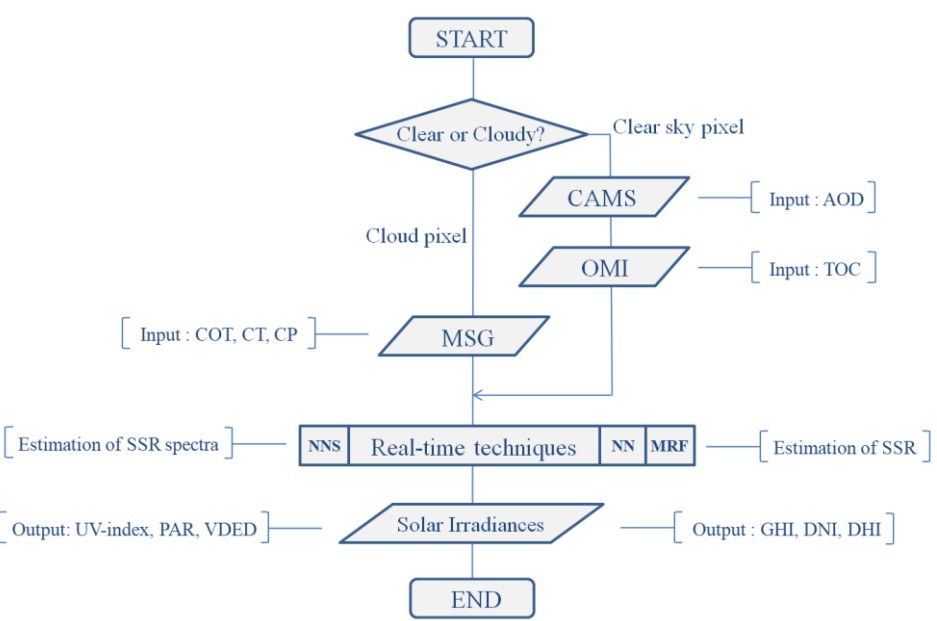

**Figure 1:** Flowchart illustration of the modelling techniques scheme. The initial pixel classification followed by the clear or cloudy sky inputs to the real-time solver result the spectral (NNS) and integrated (MRF and NN) SSR-related outputs.



**Figure 2:** An example of the output maps based on the real-time SSR techniques. Here is the GHI for the 15 April 2015 at 12:00 UTC together with the BSRN station locations.



**Table 2:** RTM simulated GHI at 15-min time intervals as compared to the BSRN ground-based measurements in terms of correlation coefficient (r) and slope.

|  | GOB | IZA | TAM | CAB | CAM | CAR | CNR | LER | TOR |
|---|---|---|---|---|---|---|---|---|---|
| slope | 0.876 | 0.923 | 0.888 | 0.866 | 0.907 | 0.960 | 0.961 | 0.897 | 0.999 |
| r | 0.943 | 0.941 | 0.942 | 0.931 | 0.938 | 0.939 | 0.946 | 0.932 | 0.969 |

35

40





10  **Table 3:** Values of parameters used for the polynomial function (3) of the MRF technique for GHI calculations under clear sky and cloudy sky conditions.

| | $GHI_{cloudy}$ | $GHI_{clear}$ |
|---|---|---|
| $P_{00}$ | -1.049 | -0.002704 |
| $P_{10}$ | -0.0287 | -0.0944 |
| $P_{01}$ | 9.69 | 0.02856 |
| $P_{20}$ | 0.004734 | $-1.75 \cdot 10^{-16}$ |
| $P_{11}$ | -0.4306 | 0.2201 |
| $P_{02}$ | -38.08 | -0.09251 |
| $P_{30}$ | -0.0002324 | $1.115 \times 10^{-16}$ |
| $P_{21}$ | 0.008734 | $4.06 \times 10^{-16}$ |
| $P_{12}$ | 0.9871 | -0.2182 |
| $P_{03}$ | 70.37 | 0.1163 |
| $P_{40}$ | $3.59 \times 10^{-6}$ | $-4.5 \times 10^{-16}$ |
| $P_{31}$ | $4.72 \times 10^{-5}$ | $4.78 \cdot 10^{-16}$ |
| $P_{22}$ | -0.01637 | 0.08 |
| $P_{13}$ | 0.9141 | -0.0498 |
| $P_{04}$ | -60.49 | -0.0132 |
| $P_{41}$ | $-2.9 \times 10^{-6}$ | $-1.2 \times 10^{-6}$ |
| $P_{32}$ | 0.0001225 | 0.001984 |
| $P_{23}$ | 0.005585 | 0.00439 |
| $P_{14}$ | 0.3199 | 0.391 |
| $P_{05}$ | 19.58 | 0.0041 |





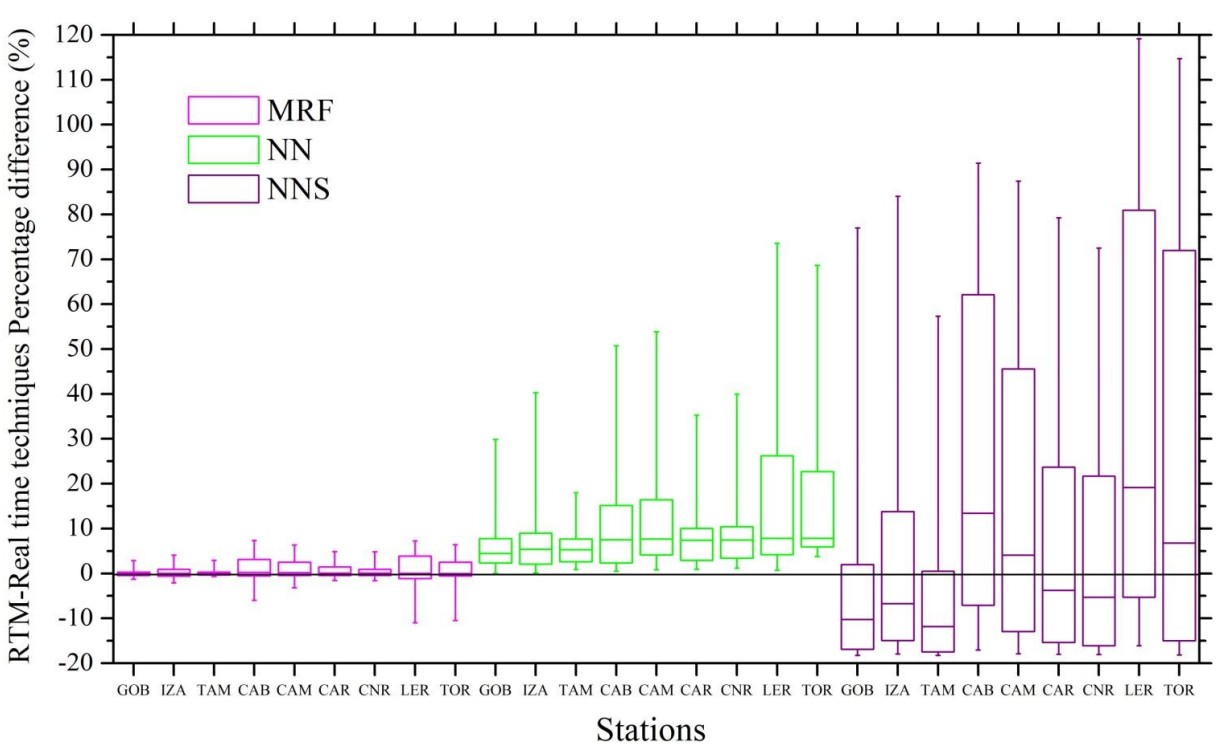

**Figure 3:** Percentage difference (%) of the real-time modelling techniques as compared to the RTM simulations for all ground stations. The box charts highlight the more precise estimation approach of the MRF technique as compared to the NN-based techniques.





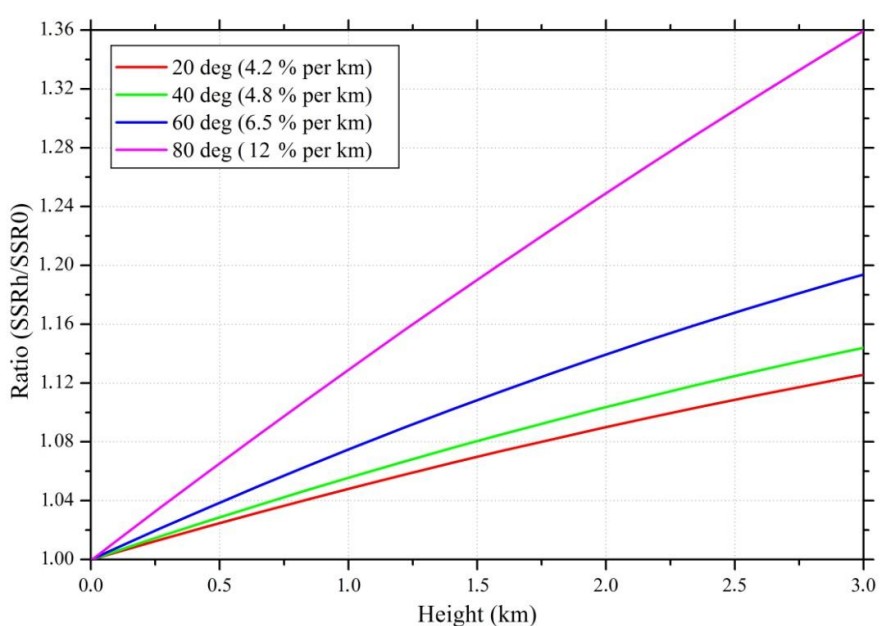

**Figure 4:** The altitude correction of GHI for various SZAs as a function of the SSR ratio (SSR at height h as to SSR at sea level).





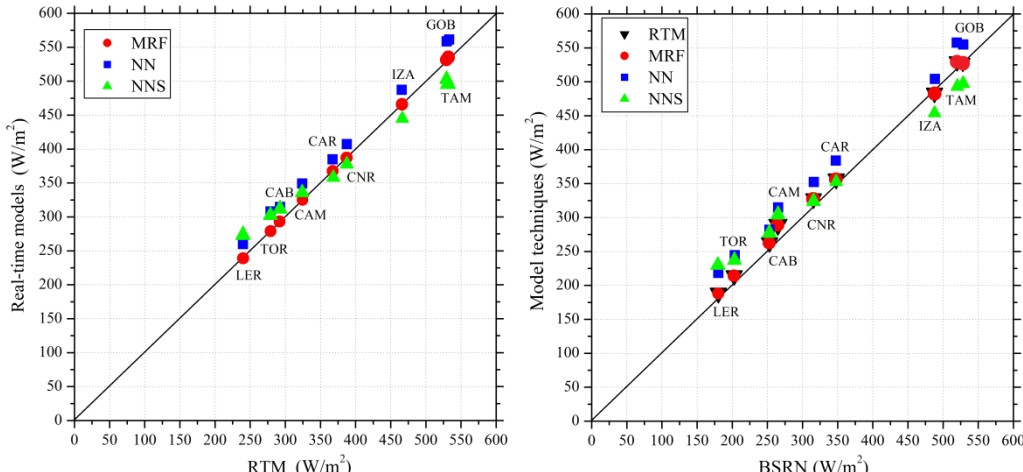

**Figure 5:** The mean GHI in W/m$^2$ of the real-time modelling techniques as compared to the RTM simulations for all ground stations (left), and the mean GHI of all models as compared to the BSRN measurements (right).

35





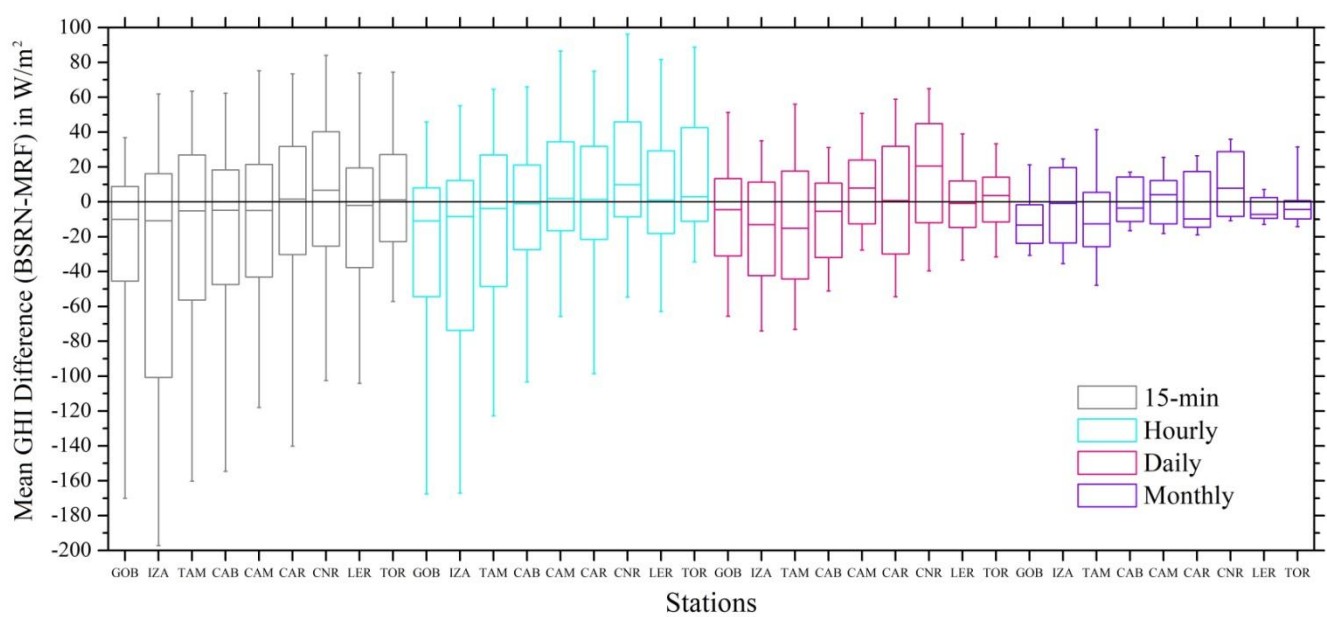

**Figure 6:** Mean GHI differences in W/m$^2$ derived by MRF as compared to the BSRN stations for each time horizon. The boxes represent
the 25$^{th}$ and 75$^{th}$ percentiles, while the in-box lines represent the median of the difference of each station. The upper and lower whiskers
represent the minimum and maximum error values.




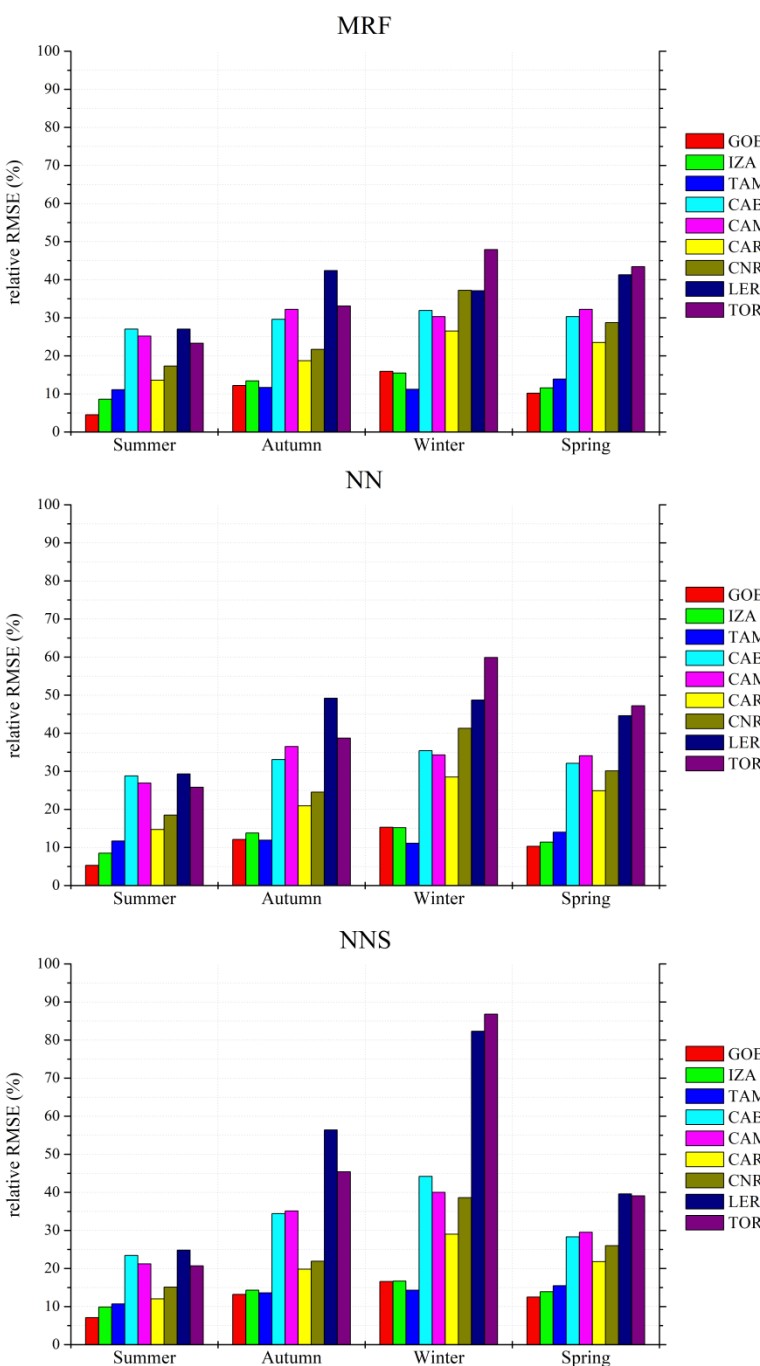

**Figure 7:** Seasonal relative RMSE values of the GHI estimations produced by the real time techniques as compared to the BSRN measurements.





**Table 4:** GHI evaluation results as a function of season and real-time techniques for all stations. The model MBE and RMSE statistical scores are shown in absolute units (W/m$^2$) and as relative magnitude (percentages in brackets).

| Station | Season | MBE | RMSE | MBE | RMSE | MBE | RMSE |
|---|---|---|---|---|---|---|---|
| | | MRF | | NN | | NNS | |
| | Winter | -20.6 (-3.2) | 103.5 (15.9) | -16.8 (-2.6) | 99.7 (15.3) | -23.4 (-3.6) | 107.8 (16.6) |
| | Spring | -5.8 (-1.1) | 55.7 (10.2) | -1.9 (-0.3) | 56.1 (10.3) | -11.6 (-2.1) | 68.2 (12.5) |
| GOB | Summer | 1.4 (0.3) | 21.9 (4.5) | 4.0 (0.8) | 25.9 (5.3) | -2.3 (-0.5) | 34.7 (7.1) |
| | Autumn | -6.9 (-1-1) | 76.0 (12.2) | -3.4 (-0.5) | 74.9 (12.1) | -11.8 (-1.9) | 81.6 (13.2) |
| | Annual period | -16.0 (-2.7) | 70.8 (12.0) | -9.1 (-1.5) | 69.6 (11.8) | -24.5 (-4.2) | 77.7 (13.2) |
| | Winter | -14.0 (-3.0) | 72.5 (15.5) | -10.5 (-2.3) | 70.9 (15.2) | -13.7 (-2.9) | 77.8 (16.7) |
| | Spring | -14.0 (-2.2) | 72.4 (11.6) | -10.9 (-1.7) | 71.4 (11.4) | -19.1 (-3.0) | 86.9 (13.9) |
| IZA | Summer | -7.8 (-1.2) | 54.9 (8.6) | -5.8 (-0.9) | 54.3 (8.5) | -10.4 (-1.6) | 63.0 (9.9) |
| | Autumn | -1.4 (-0.3) | 60.8 (13.4) | 1.8 (0.4) | 62.8 (13.8) | -4.2 (-0.9) | 65.3 (14.3) |
| | Annual period | -18.6 (-3.4) | 65.6 (12.0) | -12.7 (-2.3) | 65.2 (12.0) | -23.7 (-4.4) | 73.9 (13.6) |
| | Winter | -6.3 (-1.2) | 56.7 (11.2) | -2.0 (-0.4) | 56.2 (11.1) | -11.1 (-2.2) | 71.9 (14.3) |
| | Spring | -3.5 (-0.6) | 85.4 (13.9) | 0.5 (0.1) | 85.9 (14.0) | -11.5 (-1.9) | 95.4 (15.5) |
| TAM | Summer | 5.4 (1.0) | 61.4 (11.1) | 7.7 (1.4) | 64.7 (11.7) | 1.0 (0.2) | 59.0 (10.7) |
| | Autumn | -7.0 (-1.3) | 61.4 (11.7) | -4.2 (-0.8) | 62.5 (11.9) | -10.5 (-2.0) | 71.5 (13.6) |
| | Annual period | -5.7 (-1.0) | 67.2 (12.2) | 1.0 (0.2) | 68.3 (12.4) | -16.0 (-2.9) | 75.6 (13.7) |
| | Winter | 2.9 (2.7) | 34.5 (31.9) | 5.0 (4.7) | 38.3 (35.4) | 10.3 (9.5) | 47.7 (44.2) |
| | Spring | 10.4 (3.6) | 88.1 (30.3) | 14.1 (4.9) | 93.3 (32.1) | 10.7 (3.7) | 82.2 (28.3) |
| CAB | Summer | 13.8 (4.7) | 79.8 (27.0) | 16.3 (5.5) | 85.1 (28.8) | 11.5 (3.9) | 69.1 (23.4) |
| | Autumn | 6.6 (4.5) | 44.1 (29.6) | 9.0 (6.1) | 49.4 (33.1) | 11.5 (7.7) | 51.4 (34.4) |
| | Annual period | 16.9 (7.7) | 65.7 (29.9) | 22.3 (10.1) | 70.4 (32.0) | 22.0 (10.0) | 64.1 (29.2) |
| | Winter | 4.6 (3.6) | 38.5 (30.3) | 7.6 (6.0) | 43.6 (34.3) | 11.9 (9.4) | 50.8 (40.0) |
| | Spring | 13.4 (4.6) | 93.3 (32.2) | 17.4 (6.0) | 98.8 (34.1) | 12.1 (4.2) | 85.5 (29.5) |
| CAM | Summer | 14.6 (4.5) | 82.3 (25.2) | 17.5 (5.4) | 88.2 (26.9) | 10.8 (3.3) | 69.4 (21.2) |
| | Autumn | 10.1 (6.2) | 52.2 (32.2) | 13.5 (8.3) | 59.2 (36.5) | 13.7 (8.4) | 57.0 (35.1) |
| | Annual period | 21.4 (9.2) | 70.2 (30.3) | 28.0 (12.1) | 75.7 (32.7) | 24.2 (10.5) | 67.0 (28.9) |
| | Winter | 3.3 (1.7) | 51.2 (26.5) | 6.2 (3.2) | 55.1 (28.5) | 8.7 (4.5) | 56.0 (29.0) |
| | Spring | 10.4 (2.8) | 88.0 (23.5) | 14.1 (3.8) | 93.1 (24.9) | 6.8 (1.8) | 81.5 (21.8) |
| CAR | Summer | 8.5 (2.0) | 59.5 (13.6) | 11.3 (2.6) | 64.4 (14.7) | 3.8 (0.9) | 52.8 (12.0) |
| | Autumn | 5.5 (2.2) | 47.9 (18.7) | 8.9 (3.5) | 53.4 (20.9) | 7.0 (2.7) | 50.8 (19.8) |
| | Annual period | 13.9 (4.4) | 63.6 (20.2) | 20.2 (6.4) | 68.4 (21.7) | 13.1 (4.2) | 61.5 (19.5) |
| | Winter | 10.6 (6.5) | 60.4 (37.2) | 14.2 (8.7) | 67.1 (41.3) | 14.6 (9.0) | 62.7 (38.6) |
| | Spring | 14.0 (4.0) | 100.8 (28.7) | 18.1 (5.2) | 105.5 (30.1) | 10.1 (2.9) | 91.2 (26.0) |
| CNR | Summer | 10.2 (2.4) | 73.4 (17.3) | 12.9 (3.1) | 78.1 (18.5) | 5.5 (1.3) | 64.0 (15.1) |
| | Autumn | 9.4 (3.8) | 53.7 (21.7) | 13.0 (5.3) | 60.7 (24.5) | 10.0 (4.0) | 54.1 (21.9) |
| | Annual period | 22.1 (7.5) | 74.3 (25.1) | 29.1 (9.8) | 79.7 (27.0) | 20.0 (6.8) | 69.4 (23.5) |
| | Winter | 2.8 (5.1) | 20.3 (37.1) | 5.5 (10.1) | 26.7 (48.7) | 12.1 (22.2) | 45.1 (82.3) |
| | Spring | 15.6 (7.7) | 83.8 (41.3) | 19.9 (9.8) | 90.5 (44.6) | 18.6 (9.2) | 80.4 (39.6) |
| LER | Summer | 10.9 (4.3) | 68.8 (27.0) | 13.9 (5.4) | 74.9 (29.3) | 10.6 (4.1) | 63.2 (24.8) |
| | Autumn | 5.6 (6.4) | 37.4 (42.4) | 8.2 (9.3) | 43.3 (49.2) | 12.5 (14.2) | 49.7 (56.4) |
| | Annual period | 17.5 (10.7) | 58.2 (35.7) | 23.8 (14.6) | 64.0 (39.2) | 26.9 (16.5) | 61.1 (37.5) |
| | Winter | 4.5 (8.0) | 27.0 (47.9) | 7.5 (13.3) | 33.8 (59.9) | 13.6 (24.1) | 49.0 (86.8) |
| | Spring | 21.6 (9.6) | 97.6 (43.4) | 26.9 (12.0) | 106.1 (47.2) | 21.6 (9.6) | 87.8 (39.1) |
| TOR | Summer | 12.7 (4.4) | 67.7 (23.3) | 16.0 (5.5) | 75.0 (25.8) | 10.3 (3.5) | 60.2 (20.7) |
| | Autumn | 6.3 (5.7) | 36.7 (33.1) | 9.2 (8.3) | 42.8 (38.7) | 12.4 (11.2) | 50.3 (45.4) |
| | Annual period | 22.6 (12.6) | 63.6 (35.6) | 29.7 (16.6) | 70.5 (39.5) | 28.9 (16.2) | 63.7 (35.7) |



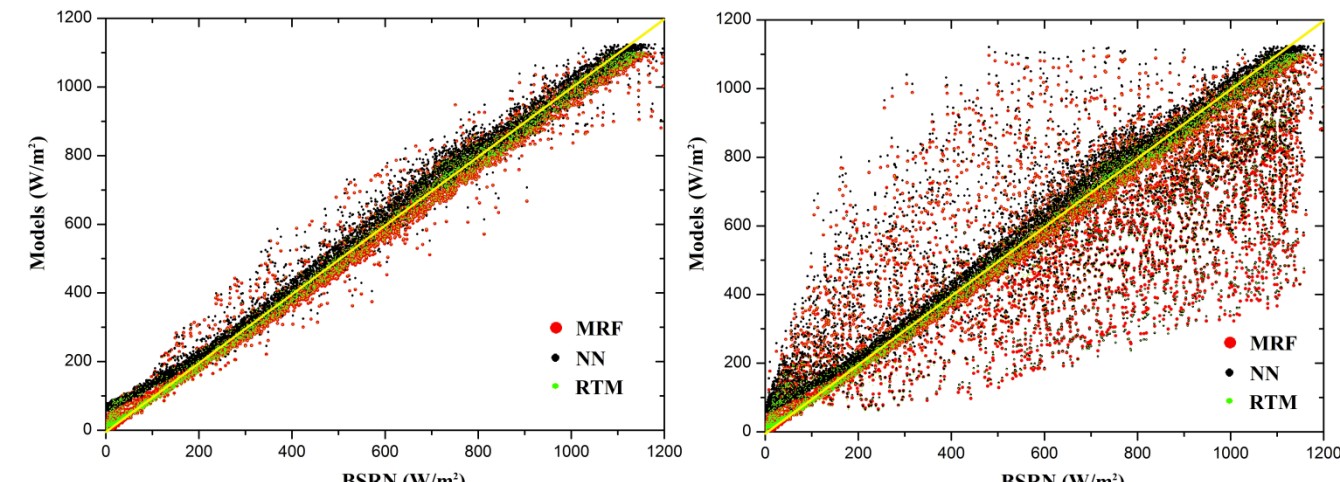

**Figure 8:** Scatterplots of real-time (MRF and NN) and RTM simulated GHI in W/m$^2$ as compared to the BSRN measurements for all stations under clear-sky (left) and all-sky (clear-sky and cloudy) conditions.





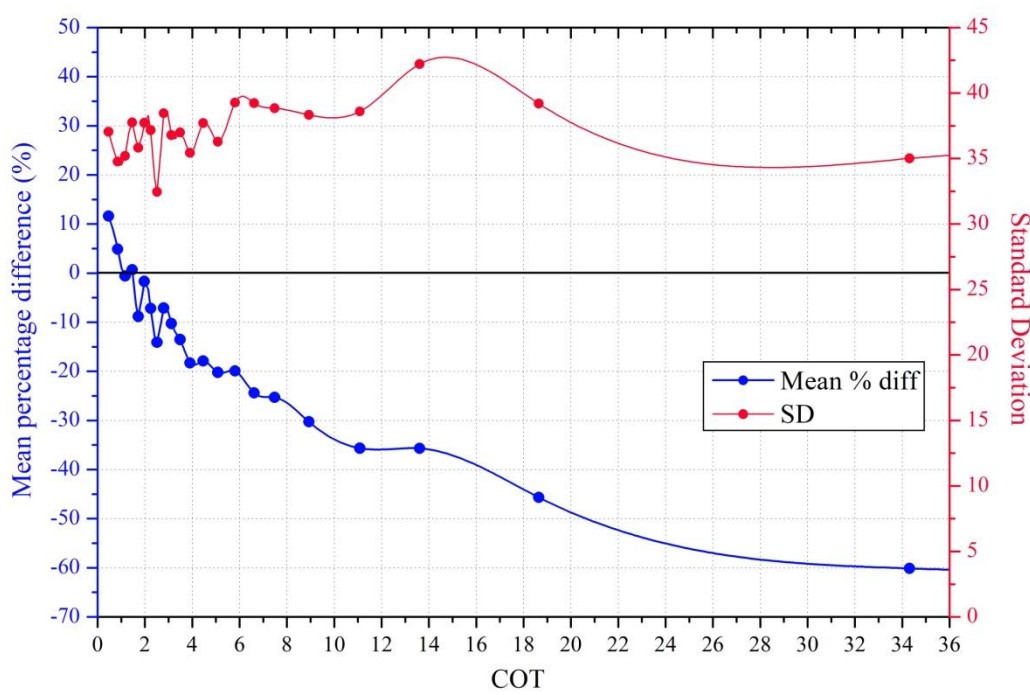

**Figure 9:** Mean percentage difference (blue) and standard deviation (red) of the 15-min GHI produced from the MRF technique as compared to ground-based measurements from all stations as a function of the COT.





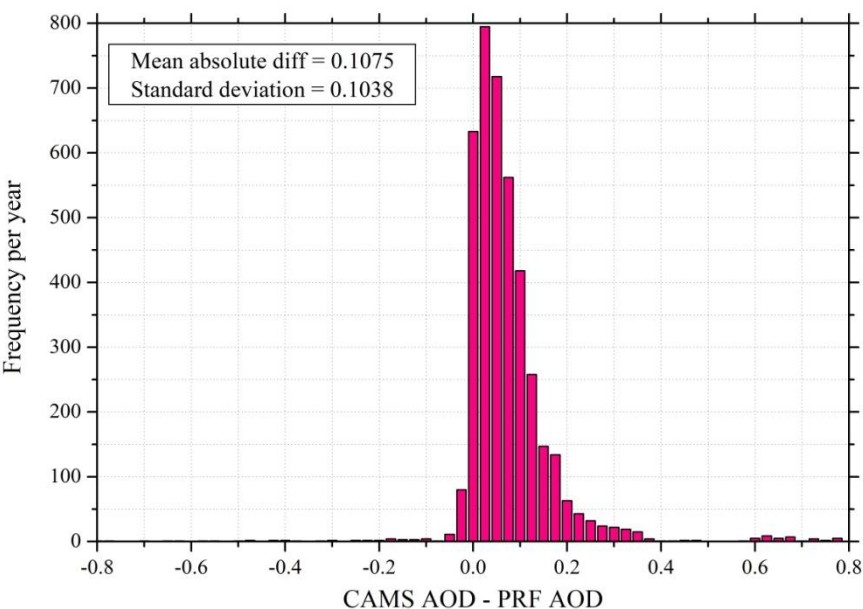

**Figure 10:** Frequency histogram of differences between the CAMS and the PFR AOD at the Izaña station together with the mean absolute
20 difference and standard deviation metrics.



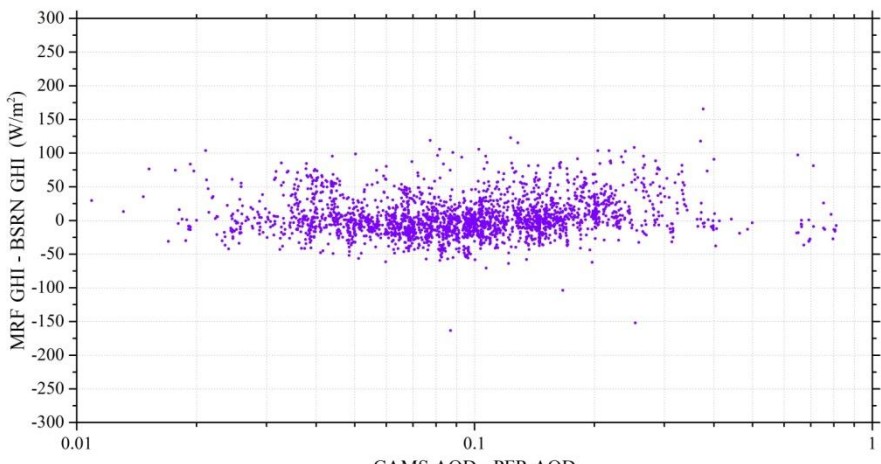

**Figure 11:** Absolute differences in GHI (in W/m$^2$) derived by the MRF technique from the ground-based measurements at Izaña (BSRN pyranometer), as a function of differences in AOD from CAMS and PFR.

