# Peer review of "Assessment of surface solar irradiance derived from real-time modelling techniques and verification with ground-based measurements"

_Atmospheric Measurement Techniques, 2017_

## Referee Comment (RC1) · Anonymous Referee #1 · 15 Nov 2017

Review of the paper:

**Title**
Assessment of surface solar irradiance derived from real-time modelling techniques and verification with ground-based measurements by Kosmopulos et al.

**General comment**
This paper shows the assessment of surface solar irradiance estimated in real time from a multi-regression function (MRF) and a neural network (NN). In the case of NN the results for the integrated spectrum and for the spectral irradiance are considered.
The output of MRF and NN are compared with Baseline Surface Radiation Network (BSRN) observations and with the training dataset, created by the libRadtran model. Nine stations with different climates are considered, giving to the discussion a general applicability.
The verification against the BSRN shows that the uncertainty of the real-time estimations of GHI ranges from -100 W/m$^2$ to 40 W/m$^2$ for the 15-min GHI, while the error decreases for averages computed for longer time ranges (down to -20, 20 W/m$^2$ for monthly averages). The impact of the aerosols and cloud optical thickness on the GHI is also discussed in detail.
The paper is well written and interesting and deserves publication on AMT. There are few points that I would remark to improve the quality of the paper.

**Specific comments**

The paper well discusses the performance of the methods presented for the estimation of the GHI in real-time. The advantages and the usability of the method is clear for the estimation of the GHI, as well as other quantities useful for the exploitation of solar energy, at a site where observations are not available. Nevertheless, because of the importance given to the paper to the fact that a real-time estimation of the GHI is presented, a discussion on the potential of the method for the short-term forecast should be provided in this paper. Also, a deeper discussion of the application fields of the methodology would be welcome.

At line 12 of page 7 it is written that NN is less computationally demanding than the interpolation. A quantification of this point should be given to better understand how the procedures work operationally.

Paragraph 3.2.1 Cloud effect: in the Figure 8, regression lines and determination coefficients should be included and discussed to give a quantitative estimation of the differences among the techniques.

**Minor comments**
Page 3 line 10: "coverege" -> "coverage".

At the end of section 2.2 explain how the relative components of the errors are computed.

Figure 9: Add units to standard deviation.

---

## Referee Comment (RC2) · Anonymous Referee #3 · 26 Dec 2017

General remarks:

The study presented in this paper is referring to the real time estimation of surface solar irradiance. Three different methodologies have been used and compered with ground-based measurements to check their performance under different conditions. The method of the multiregression function (MRF) and the neural network (NN) were also the integrated spectrum for the spectral irradiance is considered additionally (NNS). For both methods (MRF and NN) the outputs are compared with Baseline Surface Radiation Network (BSRN) observations and with the training dataset, created by the

libRadtran model. In the comparison, the authors make use of observations from nine stations covering different climatological conditions and latitudes, presenting nicely in the results section the potential of a general applicability of the proposed methods. Furthermore a sensitivity analysis about the impact of the aerosols and cloud optical depth on the GHI is also discussed in detail. The paper is well written and easy to be following by the reader, explaining in an appropriate way all the steps and the limitations of the methodology.

All in all the paper is of high interest and suitable for publication on AMT.

For completeness, there are few points that can be taking into account in the final version of the manuscript.

Specific comments:

The performance of the methods presented for the estimation of the GHI in real-time are well presented and discussed. The proposed methodology is useful for the estimation of the GHI and other relevant quantities at regions where ground based observations are missing.

The paper is focused to methods for real-time estimation of the GHI, thus a potential use of the method for forecasting application can be mentioned in the manuscript.

Paragraph 3.2.1 Cloud effect, line 15: The mentioned regression lines should be included in Figure 8.

Paragraph 3.2.2 Aerosol effect: The authors present only the comparison between Izaña ground AOD measurements and CAMS model AOD in order to estimate the errors produced by the AOD differences. As the AOD is of high importance parameter for the estimation of the GHI, the authors should stand in the manuscript (maybe in the conclusion part) that CAMS performance should be checked in case of the application of the methodology to different regions (e.g Eastern Mediterranean-Middle East).

Figure 9: Units are missing from Standard deviation

Figure 10: misspelling at the x-axes title. PRF AOD instead of PFR AOD

[Figure]

---

## Author Comment (AC1) · 2 Jan 2018

We thank reviewer #1 for the careful and constructive examination of our paper. Please find attached a zip archive with the following pdf files: (i) The answers to the reviewer comments (ii) A correction version of the paper.

Please also note the supplement to this comment:
https://www.atmos-meas-tech-discuss.net/amt-2017-351/amt-2017-351-AC1-supplement.zip

---

## Author Comment (AC2) · 2 Jan 2018

We acknowledge the reviewer for the meaningful comments. We have hopefully addressed all the points that were raised, and we are optimistic that after the reviewer's valuable corrections, the paper has been upgraded. Please find attached a zip archive with the following pdf files: (i) The answers to the reviewer comments, (ii) A correction version of the paper.

Please also note the supplement to this comment:

[Figure]

https://www.atmos-meas-tech-discuss.net/amt-2017-351/amt-2017-351-AC2-supplement.zip

---

## Author Response (AR2)

Reviewer #1

This paper shows the assessment of surface solar irradiance estimated in real time from a multi-regression function (MRF) and a neural network (NN). In the case of NN the results for the integrated spectrum and for the spectral irradiance are considered. The output of MRF and NN are compared with Baseline Surface Radiation Network (BSRN) observations and with the training dataset, created by the libRadtran model. Nine stations with different climates are considered, giving to the discussion a general applicability. The verification against the BSRN shows that the uncertainty of the real-time estimations of GHI ranges from -100 W/m2 to 40 W/m2 for the 15-min GHI, while the error decreases for averages computed for longer time ranges (down to -20, 20 W/m2 for monthly averages). The impact of the aerosols and cloud optical thickness on the GHI is also discussed in detail. The paper is well written and interesting and deserves publication on AMT. There are few points that I would remark to improve the quality of the paper.

- The paper well discusses the performance of the methods presented for the estimation of the GHI in real-time. The advantages and the usability of the method is clear for the estimation of the GHI, as well as other quantities useful for the exploitation of solar energy, at a site where observations are not available. Nevertheless, because of the importance given to the paper to the fact that a real-time estimation of the GHI is presented, a discussion on the potential of the method for the short-term forecast should be provided in this paper.

Author's reply: We thank the reviewer for this comment in order to provide additional information about the potential for short-term forecast. Since the proposed modelling techniques (MRF, NN and NNS) operate in real-time, the potential applicability for short-term forecasting purposes for the next few hours is feasible. To this direction, the CAMS AOD is already an operational forecast input (Benedetti et al., 2009) with accurate predictions every 1 hour even under high aerosol load conditions (Kosmopoulos et al., 2017). On the other hand, the MSG COT short-term forecasting requires the employment of a cloud motion vector analysis (e.g. Hammer et al., 1999) in high spatial and temporal resolution (5 x 5 km and 15 minutes, which is the MSG/SEVIRI resolution), in order to predict the impact of clouds on SSR for the next 2-3 hours, while under cloudless conditions the SZA and AOD are the main solar irradiance attenuators, and hence are available as input information to the models. The above description was added in the revised paper at the end of the sub-section 2.2.3.

- Also, a deeper discussion of the application fields of the methodology would be welcome.

Author's reply: We want to thank the reviewer for the opportunity given to us to describe the application fields of the methodology. Large scale, high temporal and spatial resolution EO-based assessment of the SSR seems to be an emerging market prospect (ITA, 2016). The potential application fields of the methodology proposed in this study include the production planning support on large scale solar farm projects and the efficient control of the electricity balancing and distribution (in support to the TSOs and DSOs), by incorporating the produced energy of the solar farms into the electricity grid. At the same time, SSR in different spectral regions highlight spectrally-weighted outputs like the UV-index (linked with skin cancer, eye cataract, DNA damage etc), the Vitamin D efficiency (related with pregnancy) and a number of agricultural and oceanographical related processes (plant photosynthesis, crop production, phytoplankton growth etc). As a result, the developed real-time modelling techniques are able to assist Public Authorities in energy planning policies, support the work of various scientific communities dealing with health protection, energy production and consumption and solar energy exploitation, and finally are able to enable the solar industry to better plan clean energies, its transmission and distribution, which in turn will boost the relative contribution to national portfolios. The above deeper discussion of the application fields of our proposed methodology has been added in the Methodology section 2.2.

- At line 12 of page 7 it is written that NN is less computationally demanding than the interpolation. A quantification of this point should be given to better understand how the procedures work operationally.

Author's reply: Using a test set of 1,000 RTM simulations from the developed LUT, we applied an interpolating function to adjacent/nearest-value and was found that each interpolation calculation required a time in excess (in total ≈ 21 hours) of each single run of RTM used to generate the LUT in the first place (≈ 12 hours for 1,000 RTM simulation outputs with spectral resolution of 1 nm in the range 285-2700 nm), while for the same test set, the NN needed almost 0.144 seconds to generate the 1,000 output spectra. Takenaka et al. (2011) have pointed out that the inclusion of many parameters (we incorporated 6 for the clear and 4 for the cloudy sky simulations) and small step sizes (we produced more than 2.5 million RTM simulations in total) can dramatically increase the LUT volume, while Sauer and Xu (1995) and Gasca and Sauer (2000) noted that the multidimensional nature of the dataset requires interpolation/extrapolation procedures that impact strongly on calculation speed. The above quantification was included in sub-section 2.2.3 in order to clarify how the procedures work operationally and to explain the reason why the NN is less computationally demanding than the interpolation.

- Paragraph 3.2.1 Cloud effect: in the Figure 8, regression lines and determination coefficients should be included and discussed to give a quantitative estimation of the differences among the techniques.

Author's reply: In the revised paper we included the regression lines and determination coefficients in Fig. 8, while we incorporated and discussed them in the Cloud effect sub-section 3.2.1. We thank the reviewer for this comment, since now the revised paper gives a quantitative estimation of the differences among the techniques.

- Page 3 line 10: "coverege" -> "coverage".

Author's reply: Corrected.

- At the end of section 2.2 explain how the relative components of the errors are computed.

Author's reply: Concerning the relative components of MBE and RMSE error measures, the normalization is done with respect to the mean ground measurement irradiance in the considered station and period. This explanation was added in the revised paper.

- Figure 9: Add units to standard deviation.

Author's reply: Corrected.

Authors: Once again, we thank the reviewer #1 for the constructive comments and we believe that after the proposed revisions this paper was overall upgraded.

Reviewer #2

The study presented in this paper is referring to the real time estimation of surface solar irradiance. Three different methodologies have been used and compared with ground-based
5   measurements to check their performance under different conditions. The method of the multi-regression function (MRF) and the neural network (NN) were also the integrated spectrum for the spectral irradiance is considered additionally (NNS). For both methods (MRF and NN) the outputs are compared with Baseline Surface Radiation Network (BSRN) observations and with the training dataset, created by the libRadtran model. In the comparison, the authors make use of observations from nine
10   stations covering different climatological conditions and latitudes, presenting nicely in the results section the potential of a general applicability of the proposed methods. Furthermore a sensitivity analysis about the impact of the aerosols and cloud optical depth on the GHI is also discussed in detail. The paper is well written and easy to be following by the reader, explaining in an appropriate way all the steps and the limitations of the methodology. All in all the paper is of high interest and suitable for
15   publication on AMT. For completeness, there are few points that can be taking into account in the final version of the manuscript.

- The performance of the methods presented for the estimation of the GHI in real-time are well
20   presented and discussed. The proposed methodology is useful for the estimation of the GHI and other relevant quantities at regions where ground based observations are missing. The paper is focused to methods for real-time estimation of the GHI, thus a potential use of the method for forecasting application can be mentioned in the manuscript.

25   Author's reply: Since the proposed modelling techniques (MRF, NN and NNS) operate in real-time, the potential applicability for short-term forecasting purposes for the next few hours is feasible. To this direction, the CAMS AOD is already an operational forecast input (Benedetti et al., 2009) with accurate predictions every 1 hour even under high aerosol load conditions (Kosmopoulos et al., 2017). On the other hand, the MSG COT short-term forecasting requires the employment of a cloud motion vector
30   analysis (e.g. Hammer et al., 1999) in high spatial and temporal resolution (5 x 5 km and 15 minutes, which is the MSG/SEVIRI resolution), in order to predict the impact of clouds on SSR for the next 2-3 hours, while under cloudless conditions the SZA and AOD are the main solar irradiance attenuators, and hence are available as input information to the models. The above description was added in the revised paper at the end of the sub-section 2.2.3.

- Paragraph 3.2.1 Cloud effect, line 15: The mentioned regression lines should be included in Fig. 8.

Author's reply: Corrected.

- Paragraph 3.2.2 Aerosol effect: The authors present only the comparison between Izaña ground AOD measurements and CAMS model AOD in order to estimate the errors produced by the AOD differences. As the AOD is of high importance parameter for the estimation of the GHI, the authors should stand in the manuscript (maybe in the conclusion part) that CAMS performance should be checked in case of the application of the methodology to different regions (e.g Eastern Mediterranean-Middle East).

Author's reply: We thank the reviewer for this comment. To this direction we added the following text in the Conclusions section. "The CAMS AOD performance has been tested as well under high aerosol loads (Kosmopoulos et al., 2017) in different regions (Eastern Mediterranean), showing similar results as compared with MODIS. However, its accuracy should be checked in case of application of the methodology to different regions (e.g. Middle East)."

- Figure 9: Units are missing from Standard deviation.

Author's reply: Corrected.

- Figure 10: misspelling at the x-axes title. PRF AOD instead of PFR AOD.

Author's reply: Corrected.

Authors: We thank reviewer #2 for the valuable corrections. We have hopefully addressed all the points that were raised, and we are optimistic that after the reviewer's corrections, the paper has been upgraded.

[revised manuscript text omitted]